# Germline Single-Nucleotide Polymorphism *GFI1-36N* Causes Alterations in Mitochondrial Metabolism and Leads to Increased ROS-Mediated DNA Damage in a Murine Model of Human Acute Myeloid Leukemia

**DOI:** 10.3390/biomedicines13010107

**Published:** 2025-01-05

**Authors:** Jan Vorwerk, Longlong Liu, Theresa Helene Stadler, Daria Frank, Helal Mohammed Mohammed Ahmed, Pradeep Kumar Patnana, Maxim Kebenko, Eva Dazert, Bertram Opalka, Nikolas von Bubnoff, Cyrus Khandanpour

**Affiliations:** 1Department of Hematology and Oncology, University Cancer Center Schleswig-Holstein (UCCSH), University Hospital Schleswig-Holstein, 23562 Lübeck, Germany; jan.vorwerk@uksh.de (J.V.);; 2Department of Hematology, Hemostaseology, Oncology, and Pneumology, West German Cancer Center Essen-Münster (WTZ), University Hospital Münster, 49149 Münster, Germany; 3Department of Hematology, First Affiliated Hospital, Guangzhou Medical University, Guangzhou 511436, China; 4Department of Hematology and Stem Cell Transplantation, West German Cancer Center Essen-Münster (WTZ), University Hospital Essen, 45147 Essen, Germany

**Keywords:** acute myeloid leukemia, GFI1, single-nucleotide polymorphism, metabolism, ROS

## Abstract

**Background/Objectives**: GFI1-36N represents a single-nucleotide polymorphism (SNP) of the zinc finger protein Growth Factor Independence 1 (GFI1), in which the amino acid serine (S) is replaced by asparagine (N). The presence of the *GFI1-36N* gene variant is associated with a reduced DNA repair capacity favoring myeloid leukemogenesis and leads to an inferior prognosis of acute myeloid leukemia (AML) patients. However, the underlying reasons for the reduced DNA repair capacity in *GFI1-36N* leukemic cells are largely unknown. Since we have demonstrated that GFI1 plays an active role in metabolism, in this study, we investigated whether increased levels of reactive oxygen species (ROS) could contribute to the accumulation of genetic damage in *GFI1-36N* leukemic cells. **Methods**: We pursued this question in a murine model of human AML by knocking in human *GFI1-36S* or *GFI1-36N* variant constructs into the murine *Gfi1* gene locus and retrovirally expressing *MLL-AF9* to induce AML. **Results**: Following the isolation of leukemic bone marrow cells, we were able to show that the *GFI1-36N* SNP in our model is associated with enhanced oxidative phosphorylation (OXPHOS), increased ROS levels, and results in elevated γ-H2AX levels as a marker of DNA double-strand breaks (DSBs). The use of free radical scavengers such as N-acetylcysteine (NAC) and α-tocopherol (αT) reduced ROS-induced DNA damage, particularly in *GFI1-36N* leukemic cells. **Conclusions**: We demonstrated that the *GFI1-36N* variant is associated with extensive metabolic changes that contribute to the accumulation of genetic damage.

## 1. Introduction

Acute myeloid leukemia (AML) is a malignant disease of the bone marrow (BM) arising from a differentiation block in the early stages of hematopoiesis. As AML is primarily a disease of the elderly, aggressive treatment regimens including chemotherapy and allogeneic stem cell transplantation are often no longer feasible, underlining the importance of developing personalized, targeted therapeutic approaches.

The presence of a polymorphism of the transcription factor *Growth Factor Independence 1 (GFI1)* gene could provide a possible target for specific AML therapy. GFI1 plays an essential role as a transcriptional repressor during the differentiation of myeloid and lymphoid progenitor cells. In recent years, there has been increasing evidence that GFI1 not only influences hematopoiesis as an epigenetic regulator but is also directly involved in DNA repair [1]. In the coding N-terminal region, GFI1 exhibits a single-nucleotide polymorphism (SNP) in which the amino acid serine (S) in position 36 is replaced by asparagine (N) [2]. Thus, there are two variants of the GFI1 protein: the more common variant called GFI1-36S and the less common variant called GFI1-36N. The *GFI1-36N* gene variant is present in approximately 7% of the healthy population, indicating its broad distribution. Notably, its prevalence increases to up to 15% among patients with myelodysplastic syndrome (MDS) and acute myeloid leukemia (AML), suggesting that the *GFI1-36N* variant is implicated in the pathogenesis of MDS and AML [2,3,4].

We have already discovered that the leukemia-predisposing function of the GFI1-36N protein variant is caused, among other factors, by reduced DNA repair processes. We have recently shown that murine *GFI1-36N-MLL-AF9* cells exhibited increased levels of the cell cycle propagating cyclin-dependent kinases 4 (CDK4) and 6 (CDK6) leading to a faster proliferation of *GFI1-36N* leukemic cells in vitro [5]. Furthermore, we found that the presence of the *GFI1-36N* variant impedes both homologous recombination- (HR-) and O^6^-methylguanine DNA methyltransferase- (MGMT-) directed DNA repair in leukemic cells [6]. This could provide an explanation as to why carriers of the *GFI1-36N* variant exhibit more genetic alterations.

In addition to direct deficits in DNA repair, reactive oxygen species (ROS) are also known to induce DNA damage. ROS are oxygen radicals such as superoxide anions (O_2_^−^) or hydrogen peroxide (H_2_O_2_), which are formed in the respiratory chain by NADPH oxidase-catalyzed reactions or by ionizing radiation [7,8]. The radicals can then react with numerous other molecules in the cell, such as lipids or metals, and form new radicals as part of a radical chain reaction. This induces DNA damage on the one hand and impairs DNA repair mechanisms on the other [9].

## 2. Materials and Methods

### 2.1. Laboratory Animal Strain

The laboratory mice (Mus musculus) used for the project originate from the C57BL/6 strain of the Jackson Laboratory (Bar Harbor, ME, USA). The findings described in this article refer to h*GFI1-36S* and h*GFI1-36N* mice that expressed either the human *GFI1-36S* or the human *GFI1-36N* gene variants, respectively, instead of the murine *Gfi1*. The integration of the human gene into the murine *Gfi1* locus was described earlier [10]. Mice expressing the human *GFI1* gene variants are referred to as either *GFI1-36S* or *GFI1-36N* mice. The *Gfi1*-expressing mice were purchased from Charles River Laboratories (Wilmington, MA, USA). The *GFI1-36S* and *GFI1-36N* mice were bred in the Central Animal Facilities of the University of Münster or the University of Lübeck. The mice were genotyped using polymerase chain reaction (PCR). To avoid infections and ensure high reproducibility the mice used in this study were kept under specific pathogen-free (SPF) conditions (sterilized supplies, controlled environment, health monitoring, and strict protocols for personnel) in individually ventilated cages (IVC). The cages were filled with enrichment materials, and HEPA-filtered air was circulated in a rack system. The mice were fed ad libitum with standard aseptic diets and aseptic water.

### 2.2. Generation of MLL-AF9 Leukemic Mice

In order to generate leukemic mice, they were BM transplanted with cells featuring an *MLL-AF9* fusion gene, which induces leukemogenesis and is associated with a particularly poor prognosis compared to the expression of other fusion genes such as *AML1-ETO* [11]. In the primary transplantation, lineage-negative (Lin^−^) BM cells transduced with an *MCSV-MLL-AF9-IRES-GFP* plasmid were transplanted after the mice had been lethally irradiated (7 + 3 Gy) (Appendix A). In the secondary transplantation, instead of transduced Lin^−^ cells, leukemic cells stored in liquid nitrogen were transplanted into sub-lethally irradiated (3 Gy) mice. The cells were obtained from mice previously diseased with AML in vivo. The exact procedures of primary and secondary transplantation were described previously [6]. To avoid infections after BM transplantation, enrofloxacin (Bayer, Leverkusen, GER) was added to the drinking water of the mice.

### 2.3. Collection of Leukemic Cells

Leukemic BM cells were washed out of the tibiae and femora. In addition, blood was collected from the right ventricle of the heart after a median sternotomy to perform a differential blood count using a Scil Vet abc hematology analyzer (Scil Animal Care Company, Viernheim, GER). The expression of seven antigens was determined by flow cytometry to distinguish the of AML from other hematologic malignancies and to confirm the blood analyses. The Ter119/APC-B220/PerCP and CD8a/APC-CD4/PerCP stainings were used to exclude acute lymphoblastic leukemia (ALL) or mixed phenotype acute leukemia (MPAL), to which the MLL-AF9 fusion protein predisposes as well [12]. The c-Kit/APC staining and GFP detection were performed for BM cells and heart blood. Following lysis of erythrocytes, 1 × 10^6^ BM cells and the heart blood were centrifuged (336× *g*, 5 min, 4 °C) in 1 mL FACS buffer. A 10 μL BD Fc Block (BD, Franklin Lakes, NJ, USA) was added to each sample. The samples were vortexed and incubated on ice for 4 min. This was followed by double or single staining using the following antibodies: anti-Mouse CD117 (c-Kit) APC (#105811), anti-Mouse CD4 PerCP (#100537), anti-Mouse CD8a APC (#100711), anti-Mouse Ly-6G/Ly-6C (Gr-1) APC (#108411), anti-Mouse TER-119 APC (#116211), anti-Mouse/Human CD11b PerCP/Cy 5.5 (#101229), anti-Mouse/Human CD45R/B220 PerCP (#103235) (all BioLegend, San Diego, CA, USA). These were diluted in FACS buffer to a concentration of 0.2 μg/mL. An 80 μL of the antibody solution was added to each sample and incubated on ice for 10 min. FACS analysis was performed using the Attune NxT Acoustic Focusing Cytometer running on the Attune NxT V3 software (Thermo Fisher Scientific, Waltham, MA, USA). The outcomes of detecting leukemia with the *GFI1-36S-MLL-AF9* or *GFI1-36N-MLL-AF9* cells have already been shown in previous publications and are, therefore, not included as a separate figure in this study [5,6].

### 2.4. Generation of GFI1-36N-K562 Cells

Human K562 cells were sourced from the Leibniz Institute DSMZ (Braunschweig, GER) and cultured in RPMI medium supplemented with 10% FBS and 1% penicillin-streptomycin at 37 °C in a 5% CO_2_ environment. For generation of *GFI1-36N* cells, two lentiviral plasmids were utilized. The Lenti-hyeA3A-BE4max plasmid, a modification of Lenti-117G-hyeA3A-BE4max (#157946, Addgene, Watertown, MA, USA), was combined with tet-pLKO-sgRNA-puro (#10432, Addgene). Lentiviral production and transduction followed previously described methods [13,14,15]. The infected K562 cells were selected via GFP sorting and puromycin treatment. Single-cell clones were obtained by limiting dilution in 96-well plates, expanded for 14 d, and subjected to genomic DNA sequencing, as described before [5].

### 2.5. Quantification of DNA Double-Strand Breaks (DSBs)

To test DNA repair after ROS inhibition, 250 × 10^3^ leukemic cells were treated for 48 h with either 10 mM N-acetylcysteine (NAC) or 30 μL α-tocopherol (αT). The same proportions of ddH_2_O or DMSO were added to the untreated control. To induce DNA damage, the cells were then irradiated with 3 Gy in a Multi Rad225 irradiation system (Precision X-ray, Madison, CT, USA) and incubated for 30, 60, and 120 min. Cells were centrifuged with FACS buffer (395× *g*, 4 min, 4 °C), the supernatant was removed and 100 μL BD Cytofix/Cytoperm buffer (BD, Franklin Lakes, NJ, USA) was added. Samples were vortexed and incubated on ice for 25 min. The cells were then washed in 1 mL BD Perm/Wash buffer (BD) and centrifuged (395× *g*, 4 min, 4 °C). The conjugated γ-H2AX antibody anti-Mouse H2A.X APC (#sc-517336, BioLegend) was diluted to 1:100 in FACS buffer. In total, 100 μL of the antibody dilution was added to each sample. The sample was vortexed and incubated at room temperature (RT) for 1 h protected from light. The cells were then washed twice in phosphate-buffered saline (PBS). Finally, 300 μL FACS buffer was added to each sample. The samples were subsequently analyzed in the Attune NxT Acoustic Focusing Cytometer using Attune NxT V3 software (Thermo Fisher Scientific).

### 2.6. Seahorse Experiments

Seahorse Mito Stress test and Glycolysis Stress test were performed following the same procedure and the same drug concentrations as previously described by our group [16,17].

### 2.7. Mitochondrial Membrane Potential (MMP), Mitochondrial Mass (Mito-Mass), and ROS Measurement

To assess MMP and ROS levels, 0.25 × 10^6^ cells were resuspended in prewarmed (37 °C) cell culture media and stained with either 50 nM tetramethylrhodamine methyl ester perchlorate (TMRE) (#ab113852, Abcam, Cambridge, UK) or 5 μM CellROX Deep Red (#C10422, Thermo Fisher Scientific). The cells were incubated for 20 min at 37 °C in a CO_2_ incubator, followed by direct measurement using flow cytometry. For negative control in TMRE staining, carbonyl cyanide-p-trifluoromethoxyphenylhydrazone (FCCP) (#ab120081, Abcam) was used. Mito-mass was analyzed by resuspending 0.25 × 10^6^ cells in prewarmed (37 °C) cell culture media without fetal calf serum (FCS), followed by staining with 50 nM MitoTracker Deep Red (#M22426, Thermo Fisher Scientific). After 20 min of incubation at 37 °C in a CO_2_ incubator, the cells were assessed using flow cytometry as described earlier [16,17].

### 2.8. Glucose Consumption and Lactate Secretion

Glucose consumption and lactate secretion were measured using an enzyme-based colorimetric kit (Biovision, San Francisco, CA, USA) following the manufacturer’s instructions.

### 2.9. Mass Spectrometry (MS)

MS was conducted according to the same protocol we followed before and is, therefore, not explicitly addressed in this manuscript [6].

## 3. Results

### 3.1. Altered Mitochondrial Metabolism in GFI1-36N Leukemic Cells

In previous work, our group showed that reduced *GFI1* expression in primary murine AML cells leads to an increased rate of oxidative phosphorylation (OXPHOS) in vitro as well as in vivo [17]. Since it is already known that *GFI1-36N* cells and cells exhibiting low *GFI1* expression both show deficient homologous recombination (HR) activity with unimpaired non-homologous end joining (NHEJ), the question arose as to whether similar changes in metabolism are also present in *GFI1-36N* leukemic cells [1,6].

In order to obtain a cross-entity assessment that is not only focused on AML and to obtain an initial overview of the effects of the polymorphism, we addressed this question in the human myeloid leukemia cell line K562, in which we knocked in a *GFI1-36N* variant. As expected, we found that the presence of the GFI1 protein variant leads to an increase in the basal and maximal levels of OXPHOS (Figure 1A). In addition to mitochondrial OXPHOS, ATP generation also occurs non-mitochondrially through glycolysis [18]. Consequently, we further measured the glycolysis rate using Seahorse glycostress assays. Since glycolysis releases lactate as a by-product, the surrounding medium is acidified, leading to an increase in the ECAR (extracellular acidification rate). However, similar to *GFI1*-knock-down (*GFI1*-KD) cells, K562-*GFI1-36N* cells displayed no significant changes in ECAR (Figure 1B).

Next, we investigated whether we could extend those findings obtained in the cell line to a murine model of human AML. Therefore, we tested OCR (oxygen consumption rate) and ECAR in murine *GFI1-36S-* and *GFI1-36N-MLL-AF9* cells and found both parameters upregulated in the presence of the SNP (Figure 1D,E).

In both, human K562 and murine *MLL-AF9* cells, we observed that the basal and maximal OCR/ECAR ratios were increased in the presence of the *GFI1-36N* gene variant (Figure 1C,F). In K562 cells, we further showed that the OCR/ECAR ratio in heterozygous K562-*GFI1-36N* cells (+/ki) was higher than in homozygous K562-*GFI1-36S* cells and that the ratio in homozygous K562-*GFI1-36N* cells (ki/ki) was again increased compared to heterozygous K562-*GFI1-36N* cells (Figure 1C). These virtually identical results in the human cell line and murine cells prompted us to perform further experiments with *GFI1-36N-MLL-AF9* cells.

### 3.2. Increased ROS and MMP Levels in GFI1-36N-MLL-AF9 Cells

Alterations in mitochondrial metabolism are closely linked to the formation of ROS [19]. Therefore, we analyzed ROS in our murine model of human AML using flow cytometry and were able to show that *GFI1-36N* leukemic cells displayed significantly increased ROS levels compared to *GFI1-36S-MLL-AF9* cells (Figure 2A and Appendix A). We also demonstrated that *GFI1-36N-MLL-AF9* cells exhibited increased MMP (Figure 2B and Appendix A), but that the mito-mass was—contrary to expectations—decreased (Figure 2C and Appendix A).

### 3.3. ROS-Induced DNA Damage in GFI1-36N-MLL-AF9 Cells

Apart from the increased ROS level described here, we found in previous work that cells harboring the *GFI1-36N* variant accumulate more genetic damage [6]. ROS can cause DNA DSBs and thus provide a possible explanation for the genetic instability of *GFI1-36N* cells [9]. To test whether an elevated ROS level results in an increased amount of DSBs in *GFI1-36N* leukemic cells and impairs their repair, *GFI1-36S* and *GFI1-36N* leukemic cells were treated with either the ROS inhibitor NAC or the ROS inhibitor αT. The number of DSBs was quantified by determining the γ-H2AX mean fluorescence intensity (MFI) in an unirradiated control and after irradiation.

In both leukemic *GFI1-36S* and *GFI1-36N* cells, the maximum number of DSBs was detected 30 min after irradiation. While in *GFI1-36S* leukemic cells the γ-H2AX MFI decreased at subsequent time points, it remained at a constant high level in *GFI1-36N* leukemic cells (Figure 3A,C). The γ-H2AX level was higher in the untreated *GFI1-36N* cells with a maximum MFI of 3548 ± 492 than in the untreated *GFI1-36S* cells with a maximum of 1811 ± 627 (Figure 3A,C). The γ-H2AX MFI of NAC-treated *GFI1-36N* cells was also increased compared to that of treated *GFI1-36S* cells (Figure 3A). Strikingly, 10 mM NAC was only able to reduce γ-H2AX MFI in the leukemic *GFI1-36N* cells, whereas *GFI1-36S* cells were not affected by NAC treatment. Consequently, the γ-H2AX MFI normalized to untreated cells was significantly lower in *GFI1-36N* cells than in *GFI1-36S* cells (Figure 3B). A similar pattern was seen regarding the number of *GFI1-36N*-γ-H2AX^+^ cells that could be reduced by NAC (Appendix A). The leukemic *GFI1-36S* cells treated with 30 μM αT had a higher γ-H2AX intensity than the untreated *GFI1-36S* cells at all time points (Figure 3C). The treated *GFI1-36N* cells exhibited slightly decreased γ-H2AX intensity compared to the untreated *GFI1-36N* cells (Figure 3C). After normalization of the γ-H2AX MFI of treated *GFI1-36S* and *GFI1-36N* cells to the untreated control, *GFI1-36N* leukemic cells were found to have a significantly lower γ-H2AX level (Figure 3D). However, it should be noted that the significant difference in γ-H2AX MFI between *GFI1-36S-MLL-AF9* and *GFI1-36N-MLL-AF9* cells was only due to the slightly increased MFI in *GFI1-36S* cells. Accordingly, αT did not have as pronounced an effect on the reduction of *GFI1-36N*-γ-H2AX^+^ cells as NAC (Appendix A).

### 3.4. GFI1-36N-MLL-AF9 Cells Exhibit Pro-Oncogenic Metabolic Changes

Due to the changes in mitochondrial metabolism and the fact that, as previously described, *GFI1*-KD cells show extensive alterations in metabolism, we were next interested in whether such changes could also be found in the presence of the *GFI1-36N* variant. Therefore, we screened the enrichment of different metabolites in *GFI1-36N-MLL-AF9* cells by mass spectrometry. We found the highest enrichment in the metabolites of the pentose phosphate pathway, gluconeogenesis, and the Warburg effect (Figure 4A,B), as well as in DNA repair, cell cycle progression and multiple further pathways that we already addressed in previous work (Appendix A). In line with this, we discovered that *GFI1-36N* leukemic cells exhibited increased glucose consumption and lactate secretion (Appendix A).

Due to the widespread metabolic changes and the fact that we have already shown that *GFI1*-KD cells are sensitive to metformin treatment [17], we studied the effect of metformin and three further drugs—UK5099 as mitochondrial pyruvate carrier inhibitor, etomoxir as carnitine palmitoyltransferase-1 inhibitor, and BPTES (bis-2-(5-phenylacetamido-1,3,4-thiadiazol-2-yl)ethylsulfide) as glutaminase inhibitor—in our *MLL-AF9* leukemic cells. Indeed, the *GFI1-36N-MLL-AF9* leukemic cells responded slightly more favorably to metformin treatment than the *GFI1-36S-MLL-AF9* cells. However, these differences did not reach a significant score (Figure 5). Interestingly, we also demonstrated that *GFI1-36N* leukemic cells were resistant to UK5099, etoxomir, and BPTES as tested by measuring the relative cell count and that they responded significantly poorer to treatment than *GFI1-36S-MLL-AF9* cells (Figure 5).

## 4. Discussion

ROS are highly reactive molecules that influence cellular processes in a concentration-dependent manner. At low-to-moderate levels, ROS function as signaling molecules, regulating cell proliferation, differentiation, and survival, and are vital for immune defense [20]. They also activate stress-adaptive pathways like Nrf2, enhancing antioxidant defenses [21]. At high levels, ROS induce oxidative damage to proteins, lipids, and DNA causing mutations and chronic inflammation [22,23,24]. Excess ROS contribute to aging, degenerative diseases, and oncogenesis through DNA damage and genomic instability [25,26].

We have shown here that *GFI1-36N*-expressing leukemic cells are associated with a significantly increased ROS level compared to *GFI1-36S* leukemic cells. An explanation for this observation has not yet been found—it is possible that the *GFI1-36N* gene product influences respiratory chain complexes or reduces the activity of antioxidant protective mechanisms. Examples include superoxide dismutase, catalase, peroxidases, and reductases [27].

The aminothiol NAC is a synthetic precursor of intracellular cysteine and glutathione [28]. Glutathione plays an important role in ROS elimination: after superoxide radicals have been converted into hydrogen peroxide by superoxide dismutase, the latter can be split into water and oxygen by catalase or oxidize glutathione (GSH) to glutathione disulfide (GSSG). GSSG is then reduced to glutathione in an NADPH-dependent manner by glutathione reductase. NADPH is provided by glucose-6-phosphate dehydrogenase [29]. An increased GSSG–GSH ratio, therefore, indicates a high ROS level in the cell [30]. This can be reduced by NAC. αT can also reduce ROS-mediated DNA damage by scavenging lipid peroxyl radicals and thus becoming a radical itself, but one that is resonance-stabilized [31]. The free radical can be finally eliminated and αT regenerated via the previously described GSSH–GSH system [29].

As we detected increased ROS levels in *GFI1-36N*-expressing leukemic cells, this could provide a further explanation for the increased number of mutations in the presence of the *GFI1-36N* variant. Consequently, minimizing the ROS level by NAC and αT could results in a reduction in the number of DSBs, particularly in *GFI1-36N*-expressing cells. This hypothesis was investigated by treating leukemic BM cells with NAC or αT and then quantifying the number of DSBs using γ-H2AX MFI assay. As expected, the γ-H2AX MFI of the leukemic *GFI1-36N* cells in the treated cells before and after irradiation was significantly lower than that of the untreated control. In contrast, the γ-H2AX MFI in leukemic *GFI1-36S* cells was almost unchanged. The results indicate that genetic damage could be reduced by NAC specifically in *GFI1-36N*-expressing leukemic cells. This emphasizes that the elevated ROS level in *GFI1-36N-MLL-AF9* cells contributes to the accumulation of DNA DSBs and can be attenuated by ROS inhibition. Next, we addressed the question of whether αT as another ROS inhibitor can achieve a similar effect to NAC. The γ-H2AX MFI was slightly decreased in the *GFI1-36N*-expressing cells following treatment and slightly increased in the *GFI1-36S*-expressing cells. The results confirm the previously reported NAC results that the number of DSBs in leukemic *GFI1-36N* cells can be reduced by ROS inhibition. However, the differences between *GFI1-36S* and *GFI1-36N* leukemic cells were not as pronounced as in the NAC treatment experiments. Nevertheless, the results show that the DSB-protective effect of αT appears to be more potent in *GFI1-36N* leukemic cells than in *GFI1-36S* leukemic cells.

In summary, our findings reported here demonstrate that the number of DSBs in *GFI1-36N* leukemic cells could be reduced by the radical scavengers NAC and αT. *GFI1-36S* leukemic cells were not affected by NAC and only slightly affected by αT. In addition to the reduced effectiveness of DSB repair, which we have already demonstrated in previous work [6], a higher ROS level in *GFI1-36N* leukemic cells seems to be responsible for the mutations and chromosomal aberrations in the presence of *GFI1-36N* SNP. The mechanism of why *GFI1-36N* leukemic cells show an increased ROS level could not be answered in this study. One possible explanation is offered by proteins of the forkhead box (FOXO) family, whose function is GFI1-dependent [32]. The latest data from our group also indicate that lower *GFI1* expression increases the OXPHOS rate via upregulation of the FOXO1-MYC axis [17]. FOXO increases the transcription of proteins that have antioxidant effects [33]. The activity of FOXO proteins is controlled post-translationally via phosphorylation and alkylation [34]. It is conceivable that this function is lost in *GFI1-36N-MLL-AF9* cells. The resulting loss of the antioxidant effect could explain the higher ROS level and the increased DSBs in *GFI1-36N-MLL-AF9* cells. Recent data indicate that the GFI1 paralog GFI1B regulates the mitochondrial respiratory chain during leukemogenesis [16]. This could additionally influence the development of ROS.

In addition to the increased ROS level, we found that *GFI1-36N-MLL-AF9* cells and K562-*GFI1-36N* cells exhibited increased OXPHOS. Interestingly, we have already shown *GFI1*-KD cells to display increased glycolysis and OXPHOS in vivo and ex vivo [17]. In this study, we showed that the MMP was significantly increased in the presence of the *GFI1-36N* polymorphism as in the *GFI1*-KD cells. This could suggest that the increased ROS level is caused by higher mitochondrial activity, but not by an increased number of mitochondria. Overall, these findings indicate similarities between *GFI1-36N* and *GFI1*-KD cells. However, unlike in *GFI1*-KD cells, we did not find a significantly better treatment response to metformin in the cells with the *GFI1* variant—although there was a trend towards a better response.

Against our expectations, the mito-mass in *GFI1-36N* cells was not increased but, on the contrary, reduced. One possible explanation might be that elevated ROS levels can—as described earlier—directly damage mitochondrial components, including proteins, lipids, and mitochondrial DNA, compromising their function [22]. We assume that this damage may impair the electron transport chain (ETC), causing increased electron leakage and further ROS production, establishing a self-sustaining feedback loop. Additionally, mitochondrial damage may lead to hyperpolarization of the mitochondrial membrane, reflected in an elevated MMP. Hyperpolarization could result from impaired ATP synthase activity or defects in the ETC, which cause an abnormal accumulation of protons across the inner mitochondrial membrane [35]. This state exacerbates ROS production as excess electrons prematurely react with oxygen. The observed reduction in mitochondrial mass may result from the activation of quality control mechanisms, such as mitophagy, in response to oxidative damage [36]. Damaged mitochondria are selectively degraded to maintain cellular health, but if the rate of damage surpasses the capacity for mitochondrial biogenesis, the overall mito-mass decreases [37,38]. This smaller mitochondrial population may remain highly active and dysfunctional, sustaining elevated ROS output and MMP despite their reduced numbers. However, it must be emphasized that these considerations are hypotheses that have not yet been experimentally proven, but certainly offer starting points for further research.

Because of the increased amount of ROS and OXPHOS, we screened for further metabolic alterations in the presence of the *GFI1-36N* variant and observed an upregulation of pathways such as the pentose phosphate pathway and the Warburg effect. Our data indicate that the *GFI1-36N* SNP drives a metabolic reprogramming that enhances glycolytic and anabolic processes. These findings support a role of the GFI1-36N protein variant in facilitating a metabolic state favorable for rapid cell proliferation [39], which is further corroborated by increased glucose consumption and lactate secretion. Such shifts in metabolism are often associated with aggressive cancer phenotypes by providing energy and biosynthetic precursors required for uncontrolled cell growth and survival [40]. Therefore, we performed a drug screen and observed that *GFI1-36N-MLL-AF9* cells exhibited resistance to UK5099, etomoxir, and BPTES, inhibitors of mitochondrial pyruvate transport, fatty acid oxidation, and glutaminase, respectively. This resistance suggests that *GFI1-36N* leukemic cells may employ alternative metabolic pathways to maintain their proliferation and survival when challenged with drugs targeting mitochondrial metabolism. For instance, the observed enrichment in the pentose phosphate pathway could provide an alternate source of NADPH and ribose-5-phosphate, supporting redox balance and nucleotide synthesis even when pyruvate uptake is inhibited. Such adaptive metabolic flexibility might contribute to therapeutic resistance as well as inferior prognosis and present a significant hurdle in targeting malignant neoplasms with the *GFI1-36N* variant.

Due to the strong functional similarities between *GFI1-36N* cells and *GFI1*-KD cells, it would certainly be interesting in the future to investigate whether the changes in DNA repair that have already been observed in *GFI1-36N-MLL-AF9* cells can also be detected in *GFI1*-KD cells [6]. This could provide targeted therapies for patients with a low GFI1 level, and not only those with the *GFI1-36N* SNP. On another note, there is considerable interest as to whether the upregulation of the FOXO1-MYC axis, which is regarded as the cause of increased OXPHOS in *GFI1*-KD cells, is also evident in *GFI1-36N* cells. This question could not be answered in the context of the present study, but it does offer promising approaches for further research. Moreover, it would be interesting to discover whether such changes occur in other neoplasms to which the *GFI1* variant also predisposes [2,3,4].

In summary, the results presented here emphasize that the presence of the *GFI1-36N* variant in cell lines as well as in a murine model of human AML contributes to DNA damage through alterations in mitochondrial function and metabolism and that this may provide a further explanation for the accumulation of genetic damage and the poorer prognosis of carriers of the *GFI1-36N* SNP.

## Figures and Tables

**Figure 1 biomedicines-13-00107-f001:**
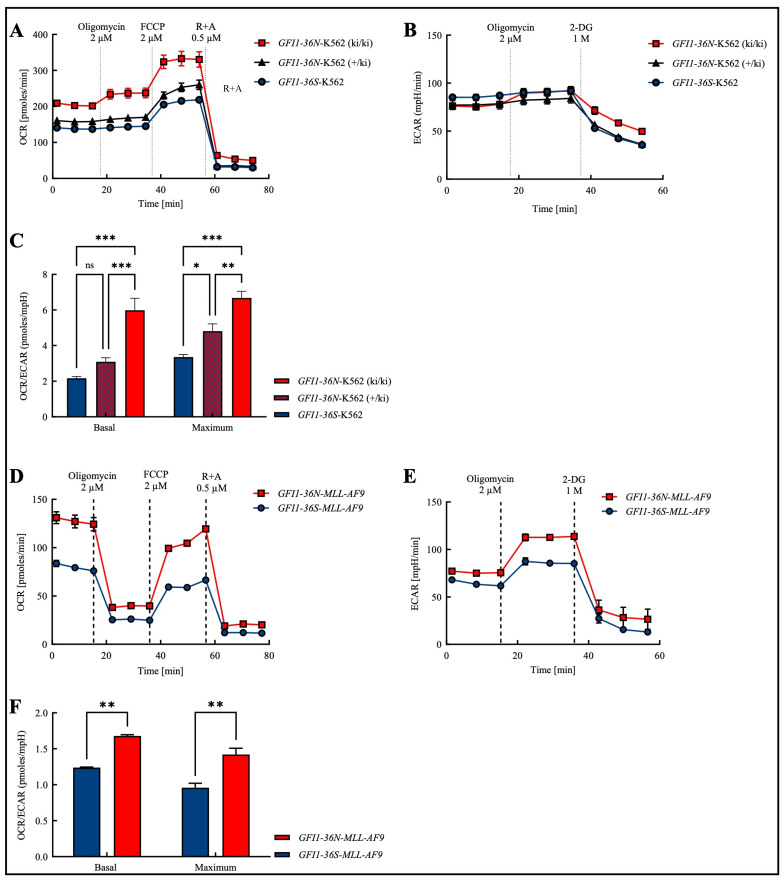
Increased OCR/ECAR ratio in the presence of the *GFI1-36N* SNP in both the human chronic myeloid leukemia (CML) cell line K562 and in the murine model of human acute myeloid leukemia (AML). (**A**): Oxygen consumption rate (OCR) was highest in *GFI1-36N*-K562 (ki/ki) cells while (**B**): extracellular acidification rate (ECAR) remained unchanged between *GFI1-36N* (ki/ki), *GFI1-36N* (+/ki), and *GFI1-36S* cells. (**C**): Increased OCR/ECAR ratio in *GFI1-36N*-K562 cells. (**D**): OCR and (**E**): ECAR were both increased in *GFI1-36N-MLL-AF9* cells compared to *GFI1-36S-MLL-AF9* controls. (**F**): Elevated OCR/ECAR ratio in *GFI1-36N-MLL-AF9* cells. FCCP: carbonyl cyanide-*p*-trifluoromethoxyphenylhydrazone; R + A: rotenone + antimycin A; 2-DG: 2-deoxyglucose. Mean ± SEM; ns: not significant; * *p* ≤ 0.05; ** *p* ≤ 0.01; *** *p* ≤ 0.001; *n* = 3–4.

**Figure 2 biomedicines-13-00107-f002:**
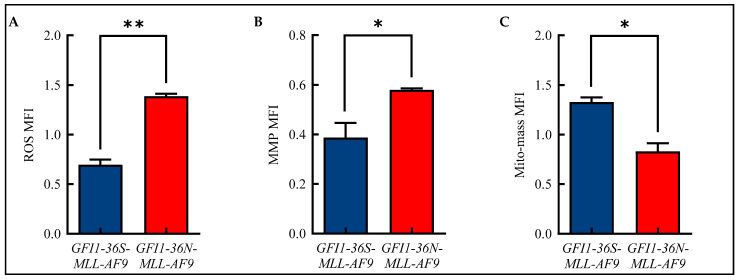
Alterations in mitochondrial metabolism. Increased (**A**): reactive oxygen species (ROS) and (**B**): mitochondrial membrane potential (MMP) with (**C**): decreased mitochondrial mass (mito-mass) in *GFI1-36N-MLL-AF9* cells. Mean ± SEM; * *p* ≤ 0.05; ** *p* ≤ 0.01; *n* = 3.

**Figure 3 biomedicines-13-00107-f003:**
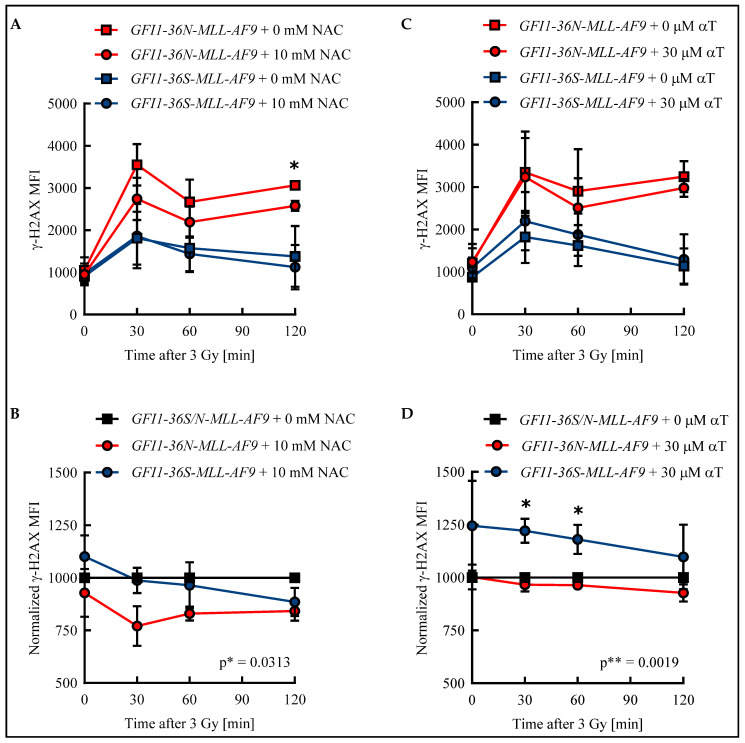
Radical scavengers N-acetylcysteine (NAC) and α-tocopherol (αT) reduced DNA damage in *GFI1-36N* leukemic cells. Leukemic bone marrow (BM) cells were incubated for 48 h with (**A**,**B**): 10 mM NAC or H_2_O or with (**C**,**D**): 30 μM αT or DMSO. After irradiation of the cells with 3 Gy, γ-H2AX MFI was determined by flow cytometry in *GFI1-36S* and *GFI1-36N* leukemic cells. B and D: After normalization to data from the untreated control, treated *GFI1-36N* leukemic cells showed fewer DNA double-strand breaks (DSBs) than *GFI1-36S* leukemic controls. Mean ± SEM; * *p* ≤ 0.05; ** *p* ≤ 0.01; *n* = 3.

**Figure 4 biomedicines-13-00107-f004:**
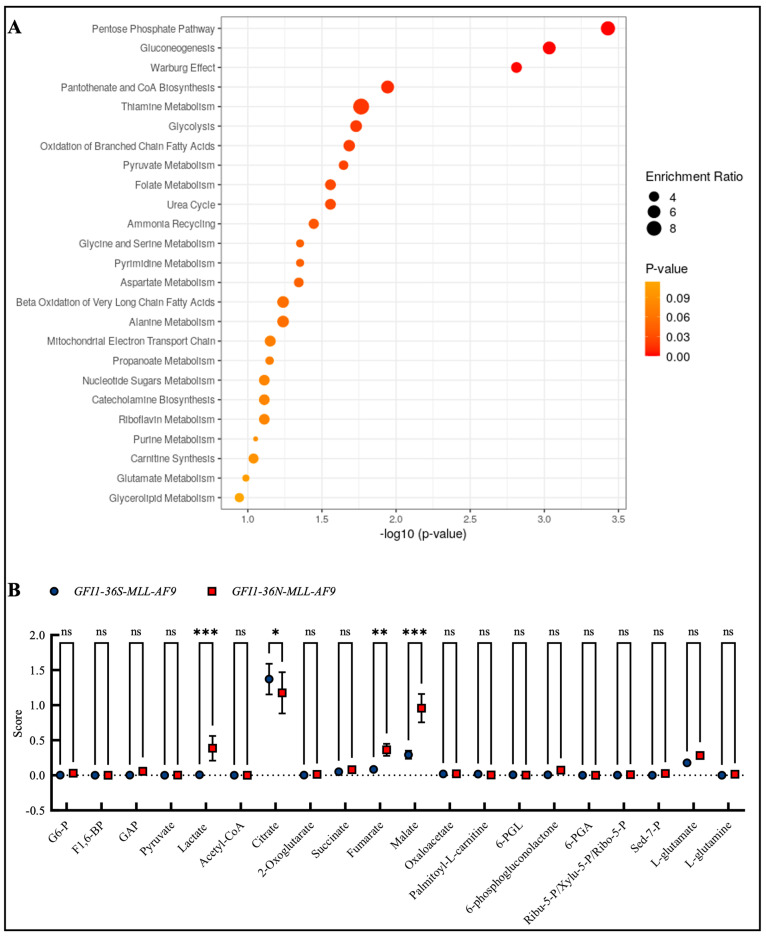
Mass spectrometry discovered enrichment of pro-oncogenic metabolites in *GFI1-36N* leukemic cells. (**A**): Top 25 upregulated metabolite sets and (**B**)**:** upregulated metabolites in *GFI1-36N-MLL-AF9* cells. Mean ± SEM; ns: not significant; * *p* ≤ 0.05; ** *p* ≤ 0.01; *** *p* ≤ 0.001; *n* = 2–3.

**Figure 5 biomedicines-13-00107-f005:**
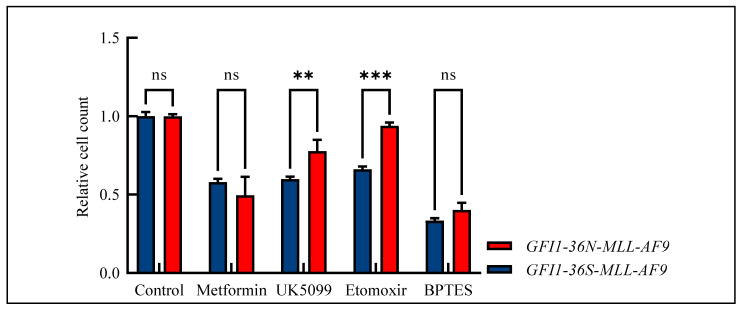
Relative cell count of *GFI1-36S-* and *GFI1-36N-MLL-AF9* cells after drug treatment normalized to control. *GFI1-36N-MLL-AF9* cells did not respond to metformin treatment and showed decreased sensitivity against treatment with UK5099, etoxomir, and BPTES. Mean ± SEM, ns: not significant; ** *p* ≤ 0.01; *** *p* ≤ 0.001; *n* = 3–6.

## Data Availability

The raw data supporting the conclusions of this article will be made available by the authors on request.

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
