# Peer review of "Germline Single-Nucleotide Polymorphism GFI1-36N Causes Alterations in Mitochondrial Metabolism and Leads to Increased ROS-Mediated DNA Damage in a Murine Model of Human Acute Myeloid Leukemia"

_biomedicines, 2025, doi:10.3390/biomedicines13010107_

Round 1

Reviewer 1 Report

Comments and Suggestions for Authors

GFI1-36N is a single-nucleotide polymorphism (SNP) of the GFI1 protein associated with leukemia. This study investigates how the GFI1-36N variant increases genetic damage in AML. There are critical issues that need to be addressed:

1. Figures 3 and 4: The results obtained from FACS are represented in a bar graph. However, the analysis data from FACS should also be provided to support the findings.

2. Mitochondrial Mass and ROS/MMP: In Figure 3, it is illogical that mitochondrial mass is decreased while ROS and MMP are increased. This discrepancy needs to be clarified.

3. Figure 4a and 4c: There is little difference between the groups even when ROS inhibitors were co-treated. This raises questions about the reliability of the claim that the expression of gamma-H2AX is mediated by ROS.

4. Figure 5: The indicators presented are known to increase due to mitochondrial metabolism. The result that these substances increase despite a decrease in mitochondrial mass is contradictory and needs further explanation.

Author Response

Dear reviewer, thank you very much for your careful study of our manuscript and for the numerous valuable suggestions for improvement. We have addressed each of your comments in detail. You will find the comments in the attached document under “Reviewer 1” from page 3.

Reviewer 2 Report

Comments and Suggestions for Authors

1. In the Abstract, proportion must be observed in the amount of the Background and the remaining sections, as the Background is too long.

2. The Methods section in the Abstract is descriptive but lacks integration. I recommend avoiding stating data in sequential lists and instead summarizing the experimental strategies.

3. In the Introduction, line 62, "GFI1-36N protein may play a role in the pathogenesis of MDS and AML" could be rewritten. I suggest "the GFI1-36N variant has been implicated in the pathogenesis of MDS and AML".

4. Please state the reason for using specific pathogen-free conditions for mice.

5. In the Discussion, I suggest providing brief data regarding the influences of ROS on other cellular processes.

Comments on the Quality of English Language

6. The whole manuscript should be rechecked as some minor mistakes are present.

Author Response

Dear reviewer, thank you very much for your careful study of our manuscript and for the numerous valuable suggestions for improvement. We have addressed each of your comments in detail. You will find the comments in the attached document under “Reviewer 2” from page 7.

Reviewer 3 Report

Comments and Suggestions for Authors

In this article, the researchers delve into the impact of the transcription factor GFI1 gene polymorphism ( GFI1-36N variant) on cellular metabolism and DNA integrity in acute myeloid leukemia (AML). The study reveal that the presence of GFI1-36N variants is linked to a notable elevation of reactive oxygen species (ROS) levels in leukemic cells. The increase of ROS may subsequently trigger an rise in DNA double-strand breaks (DSBs), ultimately influencing the onset and progression of leukemia. The study exhibits that cells harboring the GFI1-36N variant demonstrate augmented mitochondrial metabolism and glycolytic activity. Furthermore, GFI1-36N-MLL-AF9 cells exhibit pro-oncogenic metabolic changes and responded significantly poorer to treatment than GFI1-36S-MLL-AF9 cells.

The study is well designed and the paper is very well written. The discoveries underscore the potential significance of the GFI1-36N variant in the advancement of leukemia, offering a significant basis for developing targeted therapies.

However, there are certain problems that need to be resolved prior to its consideration for acceptance.

Major revision:

The specific steps involved in primary and secondary transplantation have already been elaborated in previous publication [Frank 2023]. However, the use of the same mouse image as the receptor mouse in both primary and secondary transplantation scenarios by the authors might create confusion among readers. To enhance clarity, it is advisable to revise Figure 1. Furthermore, to avoid repetition with [Frank 2023], Figure 1 can be omitted or included as a supplementary material.

Minor Revision:

Certain references cannot be found in the designated references section. Specifically, the references [Young 2020] mentioned in lines 126-127 and [Patnana 2022] in line 198 appear to be missing. It is recommended to carefully review all references for completeness.

Author Response

Dear reviewer, thank you very much for your careful study of our manuscript and for the numerous valuable suggestions for improvement. We have addressed each of your comments in detail. You will find the comments in the attached document under “Reviewer 3” from page 10.

Round 2

Reviewer 1 Report

Comments and Suggestions for Authors

All concers have been well addressed. Threre is no additional issue to raise.

Reviewer 2 Report

Comments and Suggestions for Authors

The authors addressed my concerns satisfactorily and the manuscript is appropriate enough to be published in its present form.

Reviewer 3 Report

Comments and Suggestions for Authors

The authors have addressed my all queries. I have no additional comments to provide.